# The Influence of Superabsorbent Polymers and Nanosilica on the Hydration Process and Microstructure of Cementitious Mixtures

**DOI:** 10.3390/ma13225194

**Published:** 2020-11-17

**Authors:** Gerlinde Lefever, Dimitrios G. Aggelis, Nele De Belie, Marc Raes, Tom Hauffman, Danny Van Hemelrijck, Didier Snoeck

**Affiliations:** 1Department Mechanics of Materials and Constructions, Vrije Universiteit Brussel (VUB), Pleinlaan 2, 1050 Brussels, Belgium; Dimitrios.Aggelis@vub.be (D.G.A.); Danny.Van.Hemelrijck@vub.be (D.V.H.); Didier.Snoeck@ugent.be (D.S.); 2Magnel-Vandepitte Laboratory for Structural Engineering and Building Materials, Department of Structural Engineering and Building Materials, Faculty of Engineering and Architecture, Campus A, Ghent University, Tech Lane Ghent Science Park, Technologiepark Zwijnaarde 60, B-9052 Ghent, Belgium; Nele.DeBelie@ugent.be; 3Research Group Electrochemical and Surface Engineering, Department Materials and Chemistry, Vrije Universiteit Brussel (VUB), Pleinlaan 2, 1050 Brussels, Belgium; Marc.Raes@vub.be (M.R.); Tom.Hauffman@vub.be (T.H.)

**Keywords:** superabsorbent polymer (SAP), hydrogel, nanosilica, hydration, cement, microstructure

## Abstract

Superabsorbent polymers (SAPs) are known to mitigate the development of autogenous shrinkage in cementitious mixtures with a low water-to-cement ratio. Moreover, the addition of SAPs promotes the self-healing ability of cracks. A drawback of using SAPs lies in the formation of macropores when the polymers release their absorbed water, leading to a reduction of the mechanical properties. Therefore, a supplementary material was introduced together with SAPs, being nanosilica, in order to obtain an identical compressive strength with respect to the reference material without additives. The exact cause of the similar compressive behaviour lies in the modification of the hydration process and subsequent microstructural development by both SAPs and nanosilica. Within the present study, the effect of SAPs and nanosilica on the hydration progress and the hardened properties is assessed. By means of isothermal calorimetry, the hydration kinetics were monitored. Subsequently, the quantity of hydration products formed was determined by thermogravimetric analysis and scanning electron microscopy, revealing an increased amount of hydrates for both SAP and nanosilica blends. An assessment of the pore size distribution was made using mercury intrusion porosimetry and demonstrated the increased porosity for SAP mixtures. A correlation between microstructure and the compressive strength displayed its influence on the mechanical behaviour.

## 1. Introduction

Over the last century, the concrete industry has experienced a remarkable evolution in both the design and the dimensions of built structures. To accommodate these innovations, an adaptation of the cementitious material is required, leading to the introduction of (ultra-)high-performance concrete. These types of concrete are characterized by a low water-to-cement ratio, contributing to an increased mechanical performance and a higher durability in comparison to conventional concrete. However, the limited amount of water available for hydration purpose induces self-desiccation of the cementitious matrix and leads to autogenous shrinkage. When shrinkage is restrained, cracking is likely to occur and may, in turn, affect the durability of the construction element [1,2].

A solution to mitigate autogenous shrinkage is the inclusion of superabsorbent polymers together with an additional amount of water [3]. These SAPs act as an internal curing mechanism to promote the further hydration of cement particles. During mixing, the SAPs absorb and retain part of the mixing water. In this way, an additional amount of water is present inside the mixture, without explicitly increasing the water-to-cement ratio. When self-desiccation initiates, the water is released from the SAPs and allows for continued hydration to take place. The release of water by the SAPs counteracts the self-desiccation of the cementitious matrix and reduces autogenous shrinkage [4,5,6]. Additionally, in case cracks are formed, SAPs promote the self-sealing and self-healing ability [7,8,9]. When present inside cracks, the SAPs re-absorb water or moisture from the environment, thereby physically blocking the crack entrance by their swollen volume. The release of water later on facilitates the hydration of unhydrated cement particles and the precipitation of CaCO_3_ inside the cracks.

In addition to the advantages of including SAPs into cementitious mixtures, the release of water and subsequent shrinkage of the SAPs promotes the formation of macropores. These voids affect the pore distribution and the mechanical properties of the cementitious material. The formation of cavities by the SAPs was found to create an effective air void system within the hardened matrix, similar to the use of air entraining agents. This air void system showed to be useful for increasing the freeze-thaw resistance of cementitious mixtures [10,11]. Regarding the mechanical performance, the continued hydration may lead to an increase of the compressive strength [12], but in most cases the presence of macropores reduces the mechanical performance [13,14,15]. The extent of this effect is dependent on the type of SAP and the amount used, as well as on the additional amount of water included and the water-to-cement ratio of the cementitious material [16]. Therefore, care should be taken when adding SAPs to a cementitious blend.

To improve the mechanical performance of cementitious mixtures, various additives are thoroughly studied in the literature, such as natural pozzolans and silica fume [17,18]. Over the last years, the use of nanomaterials has received increased attention. These engineered additives are able to contribute to the mechanical and durability properties in three ways [19]. Firstly, the nanoparticles act as a filler, due to their relatively small volume, leading to an increased density. Additionally, caused by their large specific surface area, the particles act as nucleation sites for the hydration of cement, promoting the formation of calcium silicate hydrates (C-S-H). Thirdly, in case pozzolanic materials such as nanoclays or nanosilica are used, an early pozzolanic reaction could occur, due to the high reactivity of the particles. The combined effect of these processes has shown to induce an increased compressive strength and a higher durability of cementitious materials [20,21]. Additionally, a fourth effect of nanoparticle inclusion can be observed, being dilution. The latter mechanism is linked to the replacement of cement by inert fillers, leading to a higher water-to-binder ratio, as the binder materials consist of a non-reactive phase. While pozzolanic nanomaterials are assumed to have an increased reactivity in comparison to the original cement, the clustering of nanoparticles may reduce this reactivity, similar to the effect of cement being partially replaced by an inert filler. The formation of agglomerates is a common issue related to the use of nanoparticles and should, therefore, be addressed in a proper way [22].

The combination of SAPs and nanosilica in cementitious mixtures was investigated in previous studies and demonstrated the mitigation of autogenous shrinkage [23] and the improvement of the self-healing ability [24], while maintaining the compressive strength of the reference material. More specifically, the addition of 0.2% of SAPs reduced the compressive strength of mortars from 77.8 ± 2.4 MPa to 70.8 ± 3.6 MPa after 90 days of curing, while the substitution of cement by nanosilica resulted in a compressive strength of 93.2 ± 2.9 MPa. The combination of both additives showed a compressive strength of 78.5 ± 1.5 MPa at the same curing age. Whereas a compensation for the negative strength reduction caused by the SAPs on the compressive performance was seen by partial replacement of cement through nanosilica, this does not imply a similar counteraction on the microstructural development. As the hardened microstructure is highly important in terms of durability of cementitious materials, the exact influence of combined SAP and nanosilica inclusion should be investigated.

Within this paper, various properties of cementitious mixtures with SAPs and nanosilica were studied. To be able to draw relevant conclusions, the additives were included both separately and jointly and a comparison to a reference mixture, without additives, was performed. The influence of SAPs and nanosilica on the hydration process was examined by means of isothermal calorimetry, measuring the heat of hydration. The initial and final setting time of mortar mixtures were determined using ultrasound, monitoring the increase in wave velocity through a cementitious sample during hardening. In the hardened state, an assessment of the pore size distribution was conducted using mercury intrusion porosimetry (MIP). This technique provides information on the total porosity and the critical pore diameter of the cementitious blends. Additionally, the amount of hydration products formed was evaluated by thermogravimetric analysis (TGA). TGA is a valuable technique to quantify the amounts of calcium hydroxide (CH) and calcium-silicate-hydrates (C-S-H) formed. Lastly, scanning electron microscopy with backscattered electrons (SEM-BSE) was used to determine the percentages of unhydrated cement, hydration products and pores on a polished cementitious specimen. A correlation between these individual results allows for an in-depth characterization of cementitious mixtures with SAPs and nanosilica, revealing the fundamental contribution of each additive to the hydration, the obtained microstructure and linked to the mechanical performance.

## 2. Materials and Methods

### 2.1. Materials

In this study, four cement pastes and their equivalent mortars were investigated, being a reference mixture without additives, a mixture including SAPs, a mixture containing nanosilica and a mixture combining SAPs and nanosilica. The Portland cement used for all blends is CEM I 52.5N Strong (Holcim, Nivelles, Belgium). The adopted water-to-binder ratio is equal to 0.35. To improve the workability of the mixtures, superplasticizer MasterGlenium 51 (conc. 35%) from BASF (Antwerp, Belgium) was included in an amount of 0.4% by weight of the binder. In case of the mortar mixture, river sand 0/2 was added in a proportion of 2:1, with respect to the binder quantity.

The superabsorbent polymer that is used is a copolymer of acrylamide and sodium acrylate, produced by bulk polymerization, with a dry particle size of 100 ± 21.5 µm and is provided by BASF (Germany). The SAP is often referred to as “SAP A” in the literature [25] and has proven its efficiency for the mitigation of autogenous shrinkage [5,26] and the improvement of the self-sealing and self-healing capacity of cementitious materials [7,8,24,27,28]. The amount of SAPs added was fixed at 0.2% of the cement mass, determined to partially mitigate autogenous shrinkage and to limit the decrease in compressive strength [23]. An additional amount of 26 g of water per gram of SAP was included on top to account for the water uptake of the SAPs. In this way, an identical workability of SAP and reference mortars was obtained. In the case of the nanosilica, a colloidal solution was used, being LUDOX^®^ HS40 (abbreviated as HS-40), provided by W. R. Grace and Co.-Conn. (Antwerp, Belgium). This colloidal dispersion contains 40 mass percentage of synthetic amorphous silica and has a density of 1.3 g/cm^3^. The nanoparticles have a nominal diameter of approximately 12 nm and a specific surface area between 198 and 258 m^2^/g. A replacement quantity of 2% by mass of cement was adopted, as this amount of nanosilica showed to compensate for the loss in strength created by the SAPs [23]. This amount concerns the dry weight of the nanosilica. As the nanosilica was present in colloidal solution, an additional volume of water was, thus, also included and was therefore subtracted from the mixing water added. To improve the dispersion of the nanoparticles, the colloidal solutions were placed in an ultrasonic bath for five minutes, prior to mixing. Moreover, as the inclusion of nanosilica lowered the workability of mortar mixtures, the amount of superplasticizer was increased to 0.76%, so that an identical flow with respect to the reference mortar was obtained. Table 1 summarizes the mixture proportions of the mortar blends. In case of cement pastes, the sand was simply omitted.

### 2.2. Isothermal Calorimetry

Using isothermal calorimetry, the heat released during the hydration of cement pastes was measured. During the experiment, samples of the considered blends are measured next to an inert reference with a similar heat capacity in a calorimeter at constant temperature. As hydration of the cementitious mixture proceeds, heat is released, while the reference sample remains at the environmental temperature. The method allows to determine the influence of various additives on the rate of hydration, which is linked to the microstructural development [29,30]. The experiments were conducted at a temperature of 20 ± 0.001 °C in a TAM-AIR isothermal calorimeter with eight channels, meaning that 16 chambers are present. The weight of the cementitious samples was equal to 20 g and as a reference material, water was taken. The hydration process was monitored for seven days.

### 2.3. Initial and Final Setting Time by Ultrasound Measurements

The initial and final setting time of cementitious mixtures are important parameters in terms of construction timings. Within this research, ultrasound was used for the determination of both setting times of mortar mixtures. This technique, based on the transmission of elastic waves, allows for continuous monitoring of the setting and hardening of cementitious mixtures. The method is based on the RILEM recommendation TC 218-SFC [31,32]. The mould is visualized in Figure 1.

From one side of the mould, a longitudinal sine wave with a frequency of 150 kHz and an amplitude of 10 V was generated and sent through the plexiglass walls and mortar material. On the other side, the signal is captured by a second acoustical sensor. The mortar was covered by means of a plastic foil to avoid evaporation and tests were conducted at a temperature of 20 ± 2 °C. Two to three replicates per mixture were made. The sensors used are of type R15α, with an operating frequency between 50 and 400 kHz and resonant frequency at 150 kHz. By monitoring the velocity of the wave signal over time, the occurrence of initial and final setting could be pinpointed. The initial setting time was determined as the first point of maximum curvature, following Reinhardt and Grosse [33]. For the final setting time, the transition between the steep increase in ultrasound velocity and the final plateau was chosen, following a study of Zhang et al. [34]. In a preliminary study, the results were compared to the outcome of the (commercially available) FreshCon setup [35] and the penetration resistance, following ASTM C 403 [36], and showed to be closely related (variation of approximately 5%). This comparison supports the application of the ultrasound method for the determination of setting times.

### 2.4. Mercury Intrusion Porosimetry

The method of mercury intrusion porosimetry (MIP) was used to evaluate the pore size distribution and total pore volume of cementitious mixtures [37,38]. During the experiment, pressurized mercury enters the accessible pores of a cementitious sample. At the start, only larger pores are filled. By increasing the pressure, smaller pores are intruded. The Washburn equation connects the diameter of a pore to the pressure that needs to be applied to enter this pore size and is given in Equation (1):(1)$$d=\frac{-4\gamma cos\theta }{P}$$

In Equation (1), *d* (µm) is the diameter of the pore, y is the surface tension equal to 0.48 J/m^2^, *θ* is the contact angle of 140° and *P* (MPa) is the absolute external pressure. The equation is however based on two assumptions, leading to some discussion whether the method is appropriate to determine pore size distributions of cementitious materials [39]. The first one approximates the pore shape as being cylindrical, which is clearly not fulfilled for cementitious mixtures. Secondly, the equation assumes equal accessibility for all pores, meaning that smaller pores are reached through the intrusion of larger pore sizes. In reality, larger pores might be connected to the outer surface by means of smaller ones. As the mercury pressure needs to be increased to access the smaller pores first, the larger pore sizes are measured as being small pores. This phenomenon is known as the ink-bottle effect [40]. While these assumptions prohibit a quantitative analysis, a qualitative comparison between different cementitious mixtures remains possible.

MIP experiments were conducted using a Pascal 140 and 440 series porosimeters from Thermo Fisher Scientific Inc. on cement pastes after 90 days of curing in plastic foil at 20 ± 2 °C. Samples of approximately 1 cm^3^ were taken and submerged in liquid nitrogen to stop the hydration reaction. Afterwards, these specimens were placed in a vacuum freeze-dryer for seven days, in order to remove any present moisture.

### 2.5. Thermogravimetric Analysis

During the thermogravimetric analysis (TGA) measurements, the weight of a sample is monitored over time while the surrounding temperature rises gradually. Due to the increase in temperature, water evaporates and phase transitions take place, which result in mass changes. This technique is often used to determine the amount of bound water, which is linked to the quantity of hydration products CH and C-S-H, and the amount of CH solely. As recommended by RILEM TC 238-SCM [41], the bound water content can be found by the mass loss between 105 °C and 650 °C, whereas the dehydroxylation of portlandite (CH) occurs between approximately 400 °C and 480 °C. By using the latter mass loss, the amount of CH can be calculated as follows:(2)$$\mathrm{CH}(\%)=\frac{{\mathrm{WL}}_{\mathrm{CH}}}{{\mathrm{WL}}_{\mathrm{final}}}\cdot \frac{{\mathrm{MW}}_{\mathrm{CH}}}{{\mathrm{MW}}_{H_{2}O}}$$
where WL_CH_ (g) is the weight loss due to dihydroxylation of CH, W (g) is the final weight of the sample and MW_CH_ (g/mol) and ${\mathrm{MW}}_{H_{2}O}$ (g/mol) are the molecular weight of CH and water, respectively. Within this study, cement paste prisms were cured for 90 days in plastic foil at 20 ± 2 °C. Afterwards, small samples of approximately 1 g were heated between 50 °C and 950 °C in a Q5000 V3.17 Build 265 calorimeter using a nitrogen atmosphere. The heating rate was fixed at 10 °C per minute.

### 2.6. Scanning Electron Microscopy—Backscattered Electrons

The method of scanning electron microscopy with backscattered electrons (SEM-BSE) is often used to examine the pore distribution of cementitious specimens in a 2D plane [42,43]. By means of a scanning electron microscope with a backscattered electron detector, black and white scans of a polished section are collected. The BSE function allows for the detection of various phases, as these appear as different grey levels. By calculating the area of each phase in the picture, an estimation of the volumetric percentage of this phase inside the cementitious blend is obtained.

Cement paste samples, originating from the same paste specimen as used for MIP and TGA, of approximately 1 cm diameter were taken. These fragments were impregnated by a low-viscosity epoxy resin, consisting of 100 g Conpox Harpiks BY 158 and 28 g of Hærder HY 2996, under vacuum conditions. Consequently, the samples were placed in an oven at 40 °C for 48 h. Afterwards, polishing of the samples was performed in five steps. During the first stage, SiC abrasive paper (320 grit) and water were used until a plane section was revealed. Secondly, SiC abrasive paper (2400 grit) and DP-lubricant brown (Struers) were employed. Additionally, three polishing steps with diamond paste and isopropanol were performed. The grain size of the diamond paste decreased per polishing stage, i.e., from 3 µm to 1 µm, and eventually to 0.25 µm. In between each polishing step, the samples were rinsed with isopropanol, dried with air and cleaned with a soft tissue. Using an end cloth at low polishing speed, the specimen was polished in the last step. Subsequently, a conductive carbon coating of 20 nm thickness was applied to the polished surface.

A JEOL JSM-IT300 device (Jeol, Peabody, MA, USA) with a BSE detector was utilized for the analysis of the polished sections. The acceleration voltage of the detector was fixed at 20 kV and a magnification of 500 was used. Per mixture design, 10–15 scans of the microstructure were taken. After collecting the black and white scans, the grey level histograms of the scans were plotted. Dark regions represent pores, while whitish zones are considered unhydrated cement. The grey levels in between these zones, signify hydration products CH and C-S-H. Figure 2 shows a SEM-BSE image and illustrates these phases. The red scale bar has a width of 50 µm. The region boundaries are adapted for every picture, to take account of the variations in contrast and brightness. The used boundaries for subsequent analysis were approximately as follows:Grey levels 0–36 represent poresGrey levels 37–200 represent CH and C-S-HGrey levels 201–255 represent unhydrated cement.

## 3. Results and Discussion

In the following sections, the hydration and microstructural development are discussed and linked to the mechanical properties. In this way, the influences of both SAPs and nanosilica next to their combined effect are highlighted and fundamentally investigated.

### 3.1. Isothermal Calorimetry

Isothermal calorimetry tests were conducted on samples of the four previously described mixtures. Cement pastes were considered instead of mortars, as the quantity of freshly mixed material needed amounts to about 20 g only and the sand does not participate to the hydration reaction. Two parameters are discussed, being the normalized heat flow (W/g) and the normalized total heat (J/g). The former is defined as the heat release per time unit and per gram of sample, while the latter is calculated as the area underneath the curve of normalized heat flow.

Using the normalized heat flow, the rate of hydration can be compared. Figure 3 depicts the normalized heat flow versus time during the first two days after mixing, as within this time period the main hydration peak occurs. The curves are shifted to account for the time between mixing and the onset of the calorimetric experiment, i.e., time zero is the moment of the first water-cement contact.

It can be seen that upon the inclusion of SAPs and extra water, the dormant period was prolonged compared to the reference cement paste. Moreover, the rate of heat release during the main hydration peak was decreased. Most likely, the absorption of alkali ions by the SAPs led to a dilution of the initial ion concentration, which decelerates the hydration reaction [44]. On the contrary, the partial substitution of cement by nanosilica HS-40 shortened the dormant period and an accelerated hydration reaction was observed. Both of these phenomena could be attributed to the nanosilica, providing an increased number of nucleation sites and showing an early pozzolanic reaction. It should be noted that the mixtures with HS-40 contain an increased amount of superplasticizer compared to reference and SAP pastes. Whereas a superplasticizer is known to delay the hydration reaction [29,45], the replacement of cement by nanosilica seemed to be substantial and cancelled out the retardation effect caused by the superplasticizer. A similar trend was seen upon comparison of SAP + HS-40 to the SAP cement paste, showing that the combined effect of the additives can actually be seen as the superposition of each material. With respect to the reference mixture, the dormant period was again shortened and an increased hydration rate was seen.

The normalized total heat, shown in Figure 4, reveals the cumulative heat released during the calorimetric experiment. The initial peak, which could be noticed in the normalized heat flow curves, was excluded from the analysis. Due to the delay between mixing and initiation of the test, part of the information of this first peak is missing, as heat is released as soon as water and cement come into contact. Therefore, the moment of minimum heat release during the dormant period is taken as the starting point.

The substitution of cement by nanosilica, accelerating the hydration reaction, can be seen by the higher cumulative heat from the start. The normalized heat reveals the contribution of SAPs and additional water to the hydration process after the main peak in heat release. Whereas the mixture with SAPs presents a longer dormant period and deceleration hydration reaction from the start, the cumulative heat curves surpass the ones of their respective references in between four to five days of curing. This feature demonstrates the continued hydration upon water release by the SAPs for internal curing, leading to an increased amount of hydration products formed (see Section 3.4).

### 3.2. Initial and Final Setting Time by Ultrasound Measurements

The determination of the initial and final setting was performed by ultrasound measurements on mortar specimens, during which the velocity of an ultrasound wave travelling through the fresh mortar was monitored over time. One representative curve of each mortar blend is shown in Figure 5. Three stages can be distinguished [35,46]. Within the first stage, the velocity stays more or less constant, as the mortar is still in the plastic state. This phase can be linked to the dormant period, seen during isothermal calorimetry. Afterwards, the velocity increases rapidly over time as soon as an interconnected network of hydration products is created. During this stage, the hydration reaction proceeds at its maximum rate, closely related to the initiation of the main peak in heat release in calorimetric experiments. Subsequently, the reaction slows down and the velocity tends to reach a plateau. The transition between the latter two stages is taken as the final setting time.

It can be seen that the dormant period is prolonged in case of SAP addition, with and without HS-40, while the reference and HS-40 curves are closely related.

Figure 6 summarizes the results of the setting times of all mixtures. In accordance with the outcome of isothermal calorimetry, the addition of SAPs delayed the initial setting compared to reference mixtures. This delay was caused by the absorption of alkali ions, diluting the ion concentration of the fresh mortar and thereby decelerating the formation of hydration products. The final setting occurred approximately at the same time as in reference mortars.

When cement was partially replaced by HS-40 together with an additional amount of superplasticizer, the initial setting time was not considerably affected. While during the calorimetric measurements an accelerated hydration reaction was observed, the initial setting time showed to be largely influenced by the superplasticizer content. This discrepancy can be explained by the characteristics of the nanosilica and the superplasticizer. The nanoparticles stimulate the formation of hydration products, increasing the amount of heat released. On the other hand, the superplasticizer repels the particles in the freshly mixed mortar, which slows down the hydration reaction but additionally prohibits the creation of a solid skeleton, i.e., the formation of links in between hydration products. The accelerated hydration reaction could however be noticed in the shortened time period between initial and final setting for HS-40 mortars, compared to the reference material.

When including both SAPs and nanosilica, the delay in initial setting time compared to the reference was again augmented, caused by the SAPs and the higher superplasticizer content compared to the reference system. The final setting time was only slightly increased, showing the effect of the nanosilica, which accelerates the hydration reaction.

### 3.3. Mercury Intrusion Porosimetry

MIP experiments were executed to determine the total porosity of cement paste mixtures and assess their pore size distribution. In Figure 7a, the total volume of intruded mercury is shown. It was seen that the reference paste presented the lowest total pore volume, equal to 65.85 mm^3^/g. When SAPs were included, only a small increase in pore volume was noticed in the case of larger pore sizes (>0.1 µm). It should be mentioned that the pores created by SAP shrinkage could not directly be measured as they are beyond the range of measurable pore sizes, having a diameter of around 250 µm. However, these pores are included in the total pore volume. The large increase in pore volume around 0.1 µm reveals the ink-bottle effect, where larger pores are filled through the intrusion of narrower pores.

Similarly, both mixtures with nanosilica presented an increased total pore volume with respect to the reference cement paste. In this case, a higher pore volume was observed for pore sizes in the µm-range, which suggests a poor compaction of the cement paste samples. On the contrary, the pore volume increased considerably at smaller pore sizes compared to the reference and SAP mixture. The derivative of the pore volume with respect to the pore diameter, depicted in Figure 7b, confirms this shift in pore size distribution. The critical pore diameter, defined as the most frequently present diameter, is much lower for HS-40 and SAP + HS-40 mixtures. This result demonstrates the influence of pore refinement by cement substitution through nanosilica and could be both attributed to the filler effect and to the pozzolanic reaction. In the case of SAP and entrained water addition, also together with HS-40, the peak in the derivative was situated closely to the one of their respective references, but was largely increased [47].

### 3.4. Thermogravimetric Analysis

TGA experiments were conducted on fractions of cement paste, originating from the same specimens as used for MIP analysis. To avoid carbonation of the samples, leading to a transformation of portlandite to CaCO_3_, the samples were tested immediately after withdrawal from the initial cement paste specimen. The weight evolution during heating is shown in Figure 8 The weight loss in between 50 °C and 105 °C is linked to the evaporation of free water. Afterwards, up to approximately 400 °C, dehydration of the cement paste specimen occurs, followed by the dehydroxylation of CH. A third steep decrease in sample weight might be observed between 680 °C and 750 °C, linked to the decarbonation of CaCO_3_ [48,49]. As no such phenomenon was observed in Figure 8, carbonation of CH did not take place.

As explained before, the portlandite content was estimated by the mass loss in between, approximately, 400 °C and 480 °C. The tangential approach was used, proposed by Scrivener et al. [49]. Secondly, the amount of bound water was taken as the mass loss between 105 °C and 650 °C. Both parameters were normalized to 100 g of anhydrous binder [41]. The results are summarized in Figure 9.

When comparing the portlandite content in a SAP cement paste to the reference, a higher amount of CH was formed. This increase is caused by the continued formation of hydration products, as the SAPs release their absorbed water. A similar behaviour was noticed during isothermal calorimetry, where an extended period of heat release demonstrated continued hydration in the presence of SAPs. Subsequently, the bound water content has also increased, due to the increased hydration and formation of CH and C-S-H.

Partial substitution of cement by nanosilica HS-40 showed to reduce the amount of portlandite, whereas a slightly elevated quantity of bound water was calculated. This phenomenon is caused by the pozzolanic nature of the nanosilica. During the pozzolanic reaction, portlandite is consumed to produce C-S-H, also seen in literature [50]. In case of combined SAP and HS-40 inclusion, an almost identical portlandite content with respect to the reference mixture was found, revealing a counteracting effect of both additives. The bound water content was higher than seen for the reference, but did not show an exact superposition of the SAP and HS-40 contribution, likely caused by the limited amount of available water.

### 3.5. Scanning Electron Microscopy—Backscattered Electrons

As explained in Section 2.6, a minimum of ten SEM-BSE scans were taken per specimen. The fractions of pores, hydration products and unhydrated cement in the cementitious blends were determined by analysing the grey level values for each phase and calculating the number of pixels corresponding to this grey level zone. A summary of the percentages of each phase is depicted in Figure 10 It should be noted that the fraction of pores was very small compared to the other phases present. The magnification used is ideal to study the capillary porosity (0.1 µm < d < 40 µm), meaning that the SAP porosity is not included. Looking at the results of MIP, indeed little intrusion was seen within this range of pore sizes.

The relatively large fraction of unhydrated cement in the reference sample, i.e., 18 ± 2%, demonstrates the low water-to-cement ratio, causing incomplete hydration of cement particles, beneficial for self-healing [24,51]. Similar to the outcome of TGA, where a higher amount of bound water was seen, the addition of SAPs led to an increase in the amount of hydration products formed and a subsequent decrease in unhydrated cement. The higher amount of hydrates was caused by the continued hydration, as the SAPs released their entrained water.

The inclusion of HS-40 had a similar effect on the percentage of formed hydrates, but this was caused by the increased availability of nucleation sites and the pozzolanic reaction of the nanosilica, during which CH is transformed to C-S-H, as could be related to the TGA experiments. The combined effect of SAP and HS-40 can be seen by the highest fraction of hydration products, together with the lowest quantity of unhydrated cement.

## 4. Conclusions

Within this study, the effect of SAPs, together with an additional amount of water, and nanosilica on the fresh and hardened properties of cementitious mixtures was investigated. The combination of both additives has already shown its benefits for the mitigation of autogenous shrinkage and the improvement of the self-healing capacity of cementitious mortars, and this while maintaining the compressive strength of the reference material. Whereas a similar behaviour under compressive loading was observed compared to the reference, the hydration progress and microstructure formation differed from one another. The information obtained from various characterization methods illustrates the combined effect of SAPs and nanosilica on various material properties and might be helpful for future modelling of the cementitious blends.

The inclusion of SAPs and additional water showed to decelerate the hydration of cement. In accordance with the lower rate of hydration, an increase in the initial setting time was observed. However, due the release of water by the SAPs, continued hydration took place, which gave rise to a higher amount of hydration products formed. This feature normally has a positive effect on the mechanical performance of cementitious materials, but was unable to counteract the influence of the macropores, created after the release of water and shrinking of the SAPs. As seen by MIP analysis, the pore volume was indeed considerably increased by the addition of SAPs, leading to a lowered compressive strength.

The use of HS-40, partially substituting cement, accelerated the hydration reaction by the early pozzolanic reaction and the increased availability of nucleation sites. Due to the presence of an additional amount of superplasticizer, the initial setting time remained comparable with respect to the reference mortar, while a much shorter final setting time was found. The filler effect, together with the pozzolanic nature of the nanoparticles, induced a pore refinement of cement pastes, meaning that a larger amount of small-sized pores was detected. In terms of hydration products formed, a decrease in the amount of CH was noticed, as CH was consumed during the pozzolanic reaction, which led to a higher quantity of C-S-H. These beneficial aspects of nanosilica confirm the increase in compressive strength compared to reference mortars.

When combining SAPs and nanosilica, a similar compressive strength in comparison to the reference material was found. Nonetheless, the hydration reaction as well as the developed microstructure showed to benefit from the inclusion of both additives. The hydration of cement was accelerated by the presence of nanosilica and a continued hydration was observed thanks to the addition of SAPs and extra water. Additionally, the amount of CH and C-S-H was increased compared to the mixture without supplements. The reason why the compressive strength was similar to the reference material is fully attributed to the macropore formation, demonstrated by the largest total porosity from MIP experiments.

## Figures and Tables

**Figure 1 materials-13-05194-f001:**
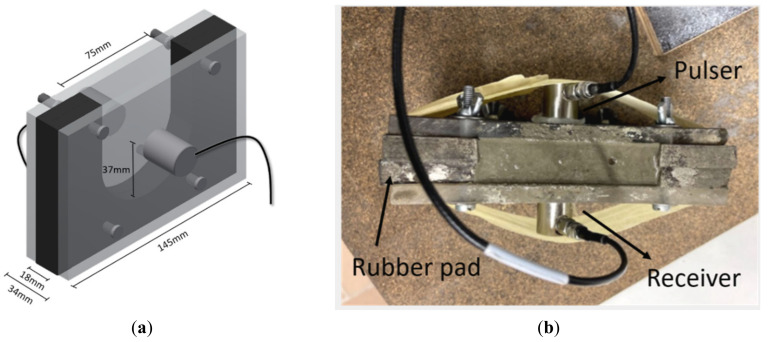
(**a**) Drawing of the ultrasound mould with acoustic sensors and (**b**) laboratory setup.

**Figure 2 materials-13-05194-f002:**
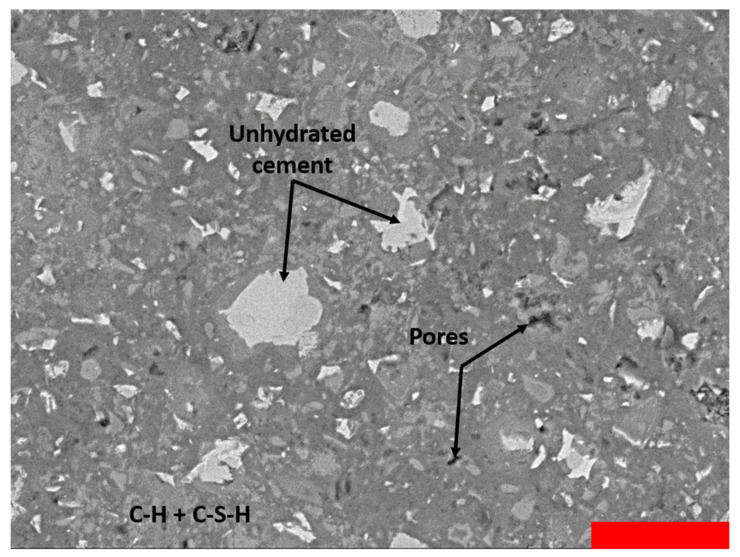
A SEM-BSE image illustrating the different phases of a cement paste. The red scale bar has a width of 50 µm.

**Figure 3 materials-13-05194-f003:**
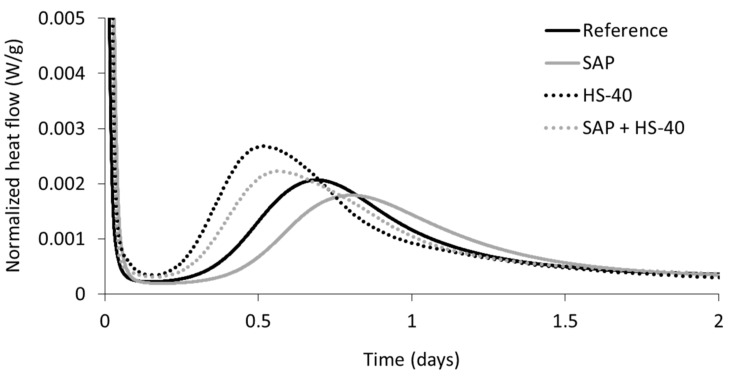
Normalized heat flow versus time of cement pastes.

**Figure 4 materials-13-05194-f004:**
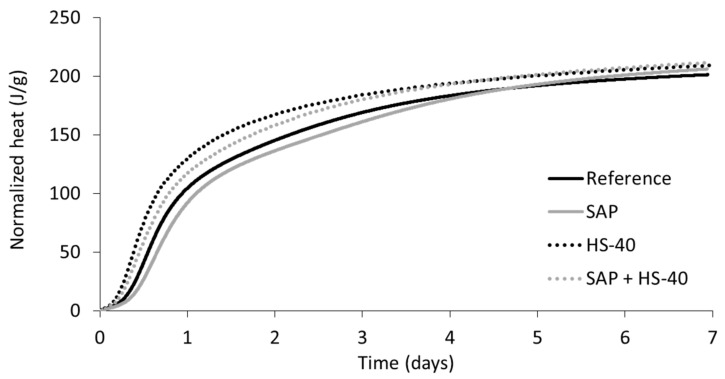
Normalized total heat versus time of cement pastes.

**Figure 5 materials-13-05194-f005:**
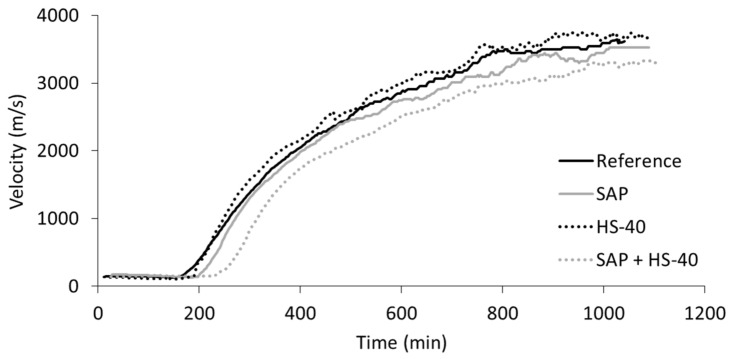
Ultrasound velocity versus time of mortar mixtures.

**Figure 6 materials-13-05194-f006:**
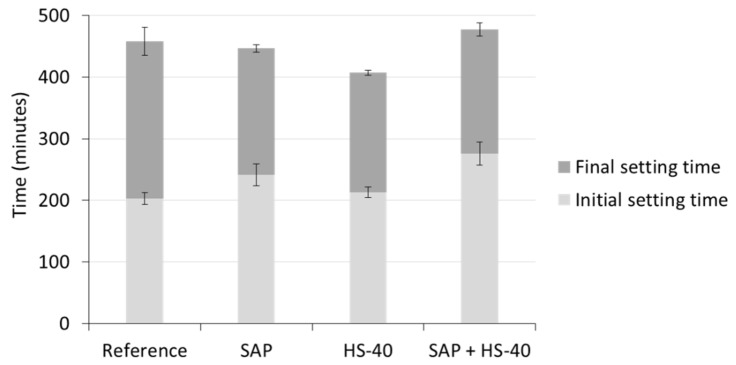
Initial and final setting by ultrasound measurements of mortar mixtures.

**Figure 7 materials-13-05194-f007:**
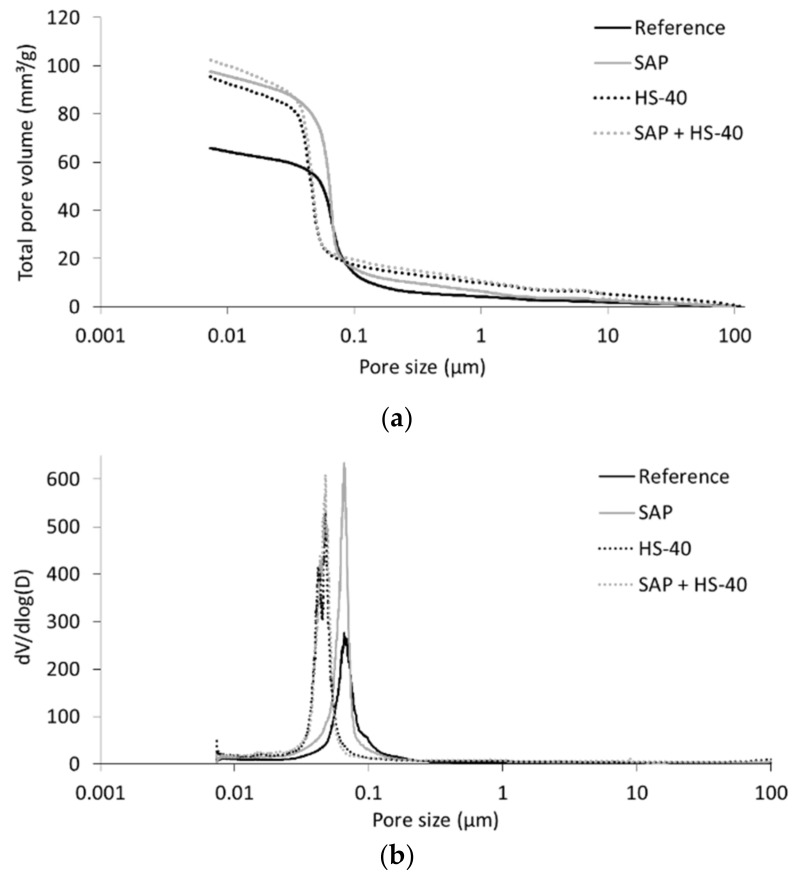
Results of MIP experiments: (**a**) Total pore volume versus pore size and (**b**) derivative of pore volume versus pore size.

**Figure 8 materials-13-05194-f008:**
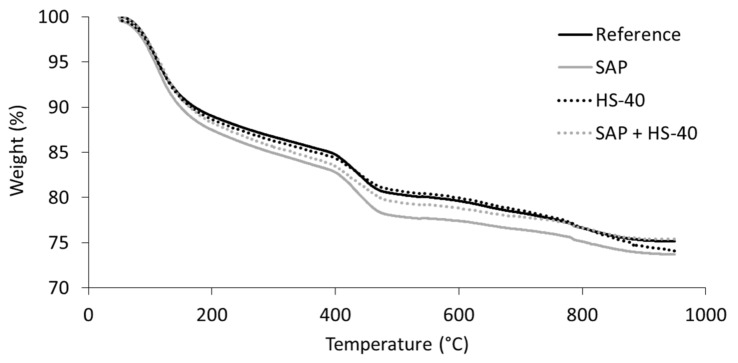
Weight versus temperature of cement pastes during TGA.

**Figure 9 materials-13-05194-f009:**
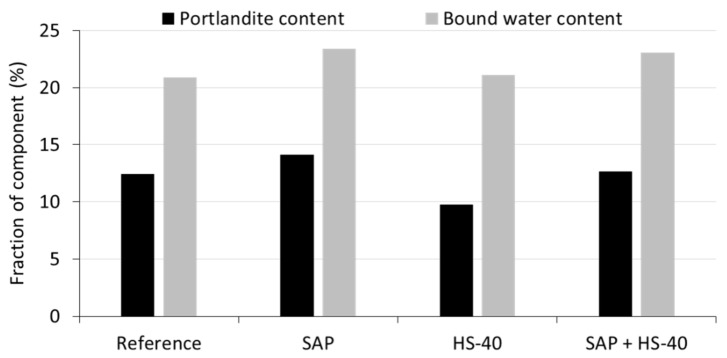
Portlandite and bound water content of cement pastes after 90 days of curing.

**Figure 10 materials-13-05194-f010:**
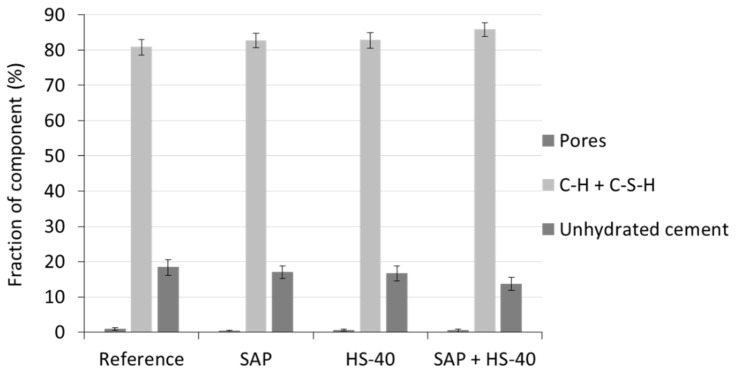
Fraction of pores, hydration products and unhydrated cement of cement pastes, determined from SEM-BSE analysis.

**Table 1 materials-13-05194-t001:** Mixture proportions of mortar mixtures (kg·m^−3^).

Mixture	Cement	Water	Sand	Superplasticizer	SAP	HS-40
Reference	580.0	203.00	1160	2.32	-	-
SAP	580.0	233.16	1160	2.32	1.16	-
HS-40	568.4	185.60	1160	4.41	-	29
SAP + HS40	568.4	215.76	1160	4.41	1.16	29

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
