# Peer review of "The Influence of Superabsorbent Polymers and Nanosilica on the Hydration Process and Microstructure of Cementitious Mixtures"

_materials, 2020, doi:10.3390/ma13225194_

Round 1

Reviewer 1 Report

This manuscript reads very well and the study on the effect of combining SAPs and nanosilica on the hydration and microstructure of cementitious mixtures is very well structured. In addition to performing a wide range of analyses and characterizations, authors successfully supported their findings by comparison with published works in the field. I recommend its publications after minor changes in the conclusion:

I suggest the authors avoid citing references in the conclusion as this part of the manuscript should contain mainly their interpretation and point of view regarding the study.

Reviewer 2 Report

  1. How is the  water absorption and release performance of SAP+HS-40?
  2.  As far as we know, SAP will release a large amount of water in the early stage of hydration due to the existence of ions. How and when does the released water affect the hydration process and the composition of hydration products?
  3.  How to ensure the uniform dispersion of SAP+HS-40?

Reviewer 3 Report

In this paper, the authors reported the influence of superabsorbent polymers and nanosilica on the hydration process and microstructure of cementitious mixtures. It as certain innovative in the research work. Take it in account and considering that in the manuscript there are some results not fully described, I recommended for publication in Materials as a complete publication after major revision.

Additional comments

1.Whole process is a routine operation. What's the innovation?

2.Only one SEM image can’t explain the phenomenon enough. Please provide photos of different scales to support your results.

3.Why authors choose LUDOX® HS40, no other sizes of the nanosilica?

4.Please quote as much as possible some literatures in recent five years.

Reviewer 4 Report

The present manuscript presents studies on various properties of cementitious mixtures with superabsorbent polymers and nanosilica. It has been previously proven that the combination of SAPs and nanosilica in cementitious mixtures is useful for the improvement of the self-healing ability [10.3390/ma13020380]. The study indicates the combined effect of SAPs and nanosilica on various material properties and possibly might be helpful for future modeling of the cementitious blends.
The submitted manuscript reports carefully well-conducted research and not raising any relevant objections. The presented results seem to be interesting for readers.

Round 2

Reviewer 3 Report

Accept in present form